# Overexpression of Aurora Kinase B Is Correlated with Diagnosis and Poor Prognosis in Hepatocellular Carcinoma

**DOI:** 10.3390/ijms25042199

**Published:** 2024-02-12

**Authors:** Jin Zhang, Jiaxiu Ma, Yalan Li, Yaxin An, Wei Du, Qun Yang, Meiling Huang, Xuefei Cai

**Affiliations:** The Key Laboratory of Molecular Biology of Infectious Diseases Designated by the Chinese Ministry of Education, Chongqing Medical University, 1 Yixue Yuan Road, Chongqing 400016, China; zhangjin0151@163.com (J.Z.); 2020111774@stu.cqmu.edu.cn (J.M.); lyl1309169225@163.com (Y.L.); a17839350271@163.com (Y.A.); duwei4728@163.com (W.D.); 15928293619@163.com (Q.Y.); 17390160045@163.com (M.H.)

**Keywords:** aurora kinase B (AURKB), hepatocellular carcinoma (HCC), cell cycle, diagnosis, prognosis

## Abstract

Aurora kinase B (AURKB) overexpression promotes tumor initiation and development by participating in the cell cycle. In this study, we focused on the mechanism of AURKB in hepatocellular carcinoma (HCC) progression and on AURKB’s value as a diagnostic and prognostic biomarker in HCC. We used data from The Cancer Genome Atlas (TCGA) and the Gene Expression Omnibus (GEO) to analyze AURKB expression in HCC. We found that the expression levels of AURKB in HCC samples were higher than those in the corresponding control group. R packages were used to analyze RNA sequencing data to identify AURKB-related differentially expressed genes (DEGs), and these genes were found to be significantly enriched during the cell cycle. The biological function of AURKB was verified, and the results showed that cell proliferation was slowed down and cells were arrested in the G2/M phase when AURKB was knocked down. AURKB overexpression resulted in significant differences in clinical symptoms, such as the clinical T stage and pathological stage. Kaplan–Meier survival analysis, Cox regression analysis, and Receiver Operating Characteristic (ROC) curve analysis suggested that AURKB overexpression has good diagnostic and prognostic potential in HCC. Therefore, AURKB may be used as a potential target for the diagnosis and cure of HCC.

## 1. Introduction

Hepatocellular carcinoma (HCC) is one of the most prevalent malignant tumors in the world, ranking sixth in terms of incidence among all cancers [1,2]. Of all liver cancer cases, hepatocellular carcinoma accounts for 90% [3,4]. Many risk factors have been found to influence the development and progression of HCC, such as liver fibrosis, cirrhosis, hepatitis B virus infection, metabolic syndrome, chronic alcohol intake, and consumption of aflatoxin-contaminated staple foods [5,6,7]. Patients with HCC can be treated with surgery (partial liver resection and liver transplantation) to improve their survival. There are also a number of other treatments available: radiation therapy, chemoembolization, and biological therapy. Although there are some methods to treat HCC, it still has a poor prognosis [8,9]. The prognosis of HCC is closely related to the stage of the disease [10]. However, most patients with HCC have already developed to the middle and advanced stages when diagnosed, and many treatments are limited, resulting in a low survival rate and frequent recurrence. Therefore, exploring more reliable early monitoring means, finding the most suitable treatment means, and grasping the best treatment opportunities are the current clinical research hotspots. If diagnosis and intervention can be carried out at an early stage, the prognosis of HCC may be improved. 

At present, alpha-fetoprotein (AFP) is the most widely used biomarker in clinics; however, its use is restricted because of its insensitivity [11]. Although there are several approaches to address this limitation, there is still an urgent need for effective, noninvasive, and highly sensitive and specific biomarkers to accurately detect early HCC [12]. In recent years, more and more studies have found that many genes can be used as biomarkers for the early screening and diagnosis of liver cancer, such as NEK2, SOX4, CTHRC1, SLC25A11, and Laminin-γ2 [13,14,15,16,17]. In addition, there are many molecules that can be used as drug targets for HCC, providing new ideas for drug treatment of HCC [18,19].

Serine/threonine protein kinases called Aurora kinases are members of the highly conserved Aurora family of mitotic kinases. Aurora kinase A, Aurora kinase B (AURKB), and Aurora kinase C are the three members of the gene family [20]. While Aurora kinase C is only expressed in the testis, Aurora kinase A and AURKB are widely expressed in most cell types [21]. AURKB plays an important role in mitosis as a chromosomal guest protein, involved in cytokinesis, the spindle checkpoint, microtubule–kinetochore attachment, chromatin remodeling, and chromosome segregation [22,23]. AURKB is primarily engaged in cell division from the G2 phase to the M phase and is located on the centromeres from the prophase through the metaphase–anaphase transition [24,25]. AURKB overexpression often leads to cell multinucleation and polyploidy, which leads to tumorigenesis, and can affect the invasion, metastasis, and drug resistance of tumor cells [26,27]. In recent years, many studies have reported that AURKB can affect the occurrence and progression of many tumors, and AURKB inhibitors can also inhibit the proliferation of cancer cells [28]. In non-small-cell lung cancer, AURKB expression may promote development. However, AURKB is a potentially poor prognostic marker for several cancers related to aneuploidy [29]. In addition, AURKB can promote the lymph node metastasis of gastric cancer and the invasion and migration of gastric cancer cells by targeting and regulating the expression of related genes [30]. AURKB expression is up-regulated in renal cell carcinoma and positively correlated with the expression of ALKBH5. ALKBH5 stabilizes AURKB mRNA in an m6A-dependent manner, thereby promoting the development of renal cell carcinoma [31]. In bladder cancer, LncRNA PVT1 regulates the invasion ability of bladder cancer cells and affects adriamycin resistance by promoting MDM2 expression and AURKB-mediated p53 ubiquitination [32]. Furthermore, high expression of AURKB is associated with poor prognosis and can predict tumor aggressiveness in HCC [29,33,34].

Although AURKB is a possible target protein and prognostic biomarker for many tumor therapies, the specific role and impact of AURKB expression on HCC progression has not been determined, especially in the diagnosis and prognosis of HCC. Therefore, we conducted in-depth bioinformatic analyses using the TCGA and GEO databases and experimentally validated the results in cells and human liver tissues to ascertain the diagnostic and prognostic significance of AURKB in HCC.

## 2. Results

### 2.1. A Higher Level of AURKB Expression Is Observed in HCC Than in Normal Tissues

Thirty-three cancer datasets acquired from the TCGA database were analyzed for AURKB expression in both tumor and normal tissues. As compared to normal tissues, AURKB was dramatically up-regulated in 18 cancer tissues, including HCC, except for cancers with missing or too few normal tissue samples (Figure 1A). AURKB expression was remarkably down-regulated in none of the examined malignancies. We examined the mRNA expression levels of AURKB in tumor and normal tissues using information from TCGA and the GEO database to determine whether they are linked to HCC. Furthermore, AURKB expression was dramatically higher (*p* < 0.001) in the HCC tissues from the GSE84402 (Figure 1B) and TCGA-HCC datasets (Figure 1C) than in normal tissues. Similarly, the analysis results of paired samples of tumor and normal tissues from the TCGA database showed that the AURKB expression level was higher in tumors (Figure 1D). Subsequently, we used clinical HCC tissue samples to detect the AURKB mRNA level, and the same result was obtained. There was a significant difference in the AURKB expression level between tumor tissues and normal tissues (Figure 1E; ** *p* < 0.01; ***, *p* < 0.001). Next, the expression of AURKB was analyzed by Western blotting from four human HCC cell lines (HepG2, PLC5, Huh7, and AD38) and two normal liver cell lines (MIHA and LO2). It is evident that almost all of the HCC cell lines examined had significantly higher levels of AURKB expression than did the normal liver cell lines (Figure 1F). Therefore, two AURKB-overexpressing HCC cell lines, PLC5 and HepG2, were selected for the following experimental verification. Immunohistochemical results were obtained from the HPA database for a total of 15 patients, including normal liver tissue from 3 patients, liver tissue from 6 patients with HCC, and liver tissue from 6 patients with cholangiocarcinoma. The patient data corresponding to the IHC samples are listed in Appendix A. Immunohistochemical results of normal liver tissues from two patients and liver tissue from two patients with HCC were selected (Figure 1G), and the results showed that AURKB was positive for expression in the HCC tissue specimens.

### 2.2. Analysis of AURKB’s Differential Expression and Functional Enrichment in HCC

Based on the median AURKB expression value, we used DEGs to separate 424 HCC patients into two groups: those with high and low AURKB expression. In the group with high AURKB expression, 289 DEGs were significantly up-regulated, and in the group with low AURKB expression, 60 DEGs were considerably down-regulated (Figure 2A). Furthermore, we used the R package “clusterProfiler” to investigate the biological mechanisms behind AURKB-associated DEGs in HCC. 

The results of Gene Set Enrichment Analysis (GSEA) showed that 16 signaling pathways were enriched in DEGs with high AURKB expression (Table 1). The GSEA results revealed that there were significant enrichment up-regulated pathways associated with mitosis, including cell cycle checkpoints (NES = 2.230, *p*.adj < 0.001, *p* value < 0.001) (Figure 2B), the resolution of sister chromatid cohesion (NES = 2.272, *p*.adj < 0.001, *p* value < 0.001) (Figure 2C), the mitotic spindle checkpoint (NES = 2.189, *p*.adj < 0.001, *p* value < 0.001) (Figure 2D), and so on.

### 2.3. AURKB Knockdown Inhibits the Proliferation of HCC Cells 

To investigate the effect of AURKB expression on HCC development and progression, we performed AURKB knockdown experiments in the HCC cell lines HepG2 and PLC5. Compared to control cells, the AURKB knockdown cell lines exhibited considerably lower levels of protein and mRNA expression (Figure 3A,B). Then, CCK-8 assays revealed that down-regulated expression of AURKB markedly suppressed the proliferation of HepG2 and PLC5 cells (Figure 3C). Later, in order to determine the impact of AURKB on the cell cycle, we carried out flow cytometry measurements of the cellular DNA content in both HepG2 and PLC5 cells. In the G2/M phase, we saw more AURKB cells than scrambled control cells; consequently, we saw fewer AURKB cells in the G1 phase of the cell cycle (Figure 3D). These findings show that HCC cell growth is inhibited in vitro by AURKB knockdown, which results in cell cycle arrest in the G2/M phase.

### 2.4. AURKB Expression Is Linked to Various Clinicopathological Characteristics in HCC

Table 2 displays the findings of our analysis examining the relationship between AURKB expression levels and clinicopathological characteristics. To examine the different clinicopathological features, 374 HCC patients in the TCGA database were divided into 187 HCC patients with low AURKB expression and 187 HCC patients with high AURKB expression according to target gene expression levels. Among them, for gender, race, clinical T stage, and pathological stage, there were differences between the low-AURKB-expression and high-AURKB-expression groups of HCC patients (*p* < 0.05). At the same time, there was a significant difference in the AFP level (*p* < 0.001) between the low-AURKB-expression and high-AURKB-expression groups of HCC patients, but no association in other clinicopathological characteristics. 

Further, the expression level of AURKB in different subgroups of these 7 clinicopathological characteristics was analyzed. There was no significant difference in AURKB expression among the gender (Figure 4A), age (Figure 4B), and tumor status (Figure 4C) subgroups, but AURKB expression was significantly higher in the T stage (Figure 4D), pathological stage (Figure 4E), AFP level (Figure 4F), and race (Figure 4G) subgroups (*p* < 0.001). Moreover, AURKB levels were increased in HCC patients with advanced disease stage and high AFP levels. AURKB expression was higher in Asian patients than in patients of other races, with significant differences between Asian and White patients (*p* < 0.01).

### 2.5. The Prognostic and Diagnostic Value of AURKB in HCC

Kaplan–Meier survival analysis and Cox regression analysis were used to examine the prognostic relevance of AURKB in HCC patients. In HCC patients, high AURKB expression was strongly correlated with poorer overall survival (OS) (*p* < 0.05) (Figure 5A), disease-specific survival (DSS) (*p* < 0.05) (Figure 5B), and progression-free interval (PFI) (*p* < 0.05) (Figure 5C), as compared to HCC patients with low AURKB expression, according to the Kaplan–Meier survival analysis. Univariate Cox regression analyses indicated that gender (male, HR = 0.793, *p* = 0.200), age (>60, HR = 1.205, *p* = 0.295), race (Black or African American, HR = 1.585, *p* = 0.290; White, HR = 1.323, *p* = 0.144), and AFP level (>400, HR = 1.075, *p* = 0.772) showed no prognostic significance, but pathologic T stage (T3 or T4, HR = 2.949, *p* < 0.001), pathologic stage (stage III or stage IV, HR = 2.823, *p* < 0.001), tumor status (with tumor, HR = 2.317, *p* < 0.001), and AURKB expression (high, HR = 1.549, *p* = 0.014) were connected to the OS of HCC patients. Multivariate Cox regression revealed that tumor status (with tumor, HR = 1.783, *p* = 0.005) and AURKB expression (high, HR = 1.561, *p* = 0.025) serve as independent risk factors for total survival in HCC patients (Table 3). A Cox regression analysis of particular HCC patient subgroups based on clinicopathological criteria was carried out to ascertain the prognostic value of AURKB, as demonstrated in Figure 5D,E.

The diagnostic value of the AURKB expression levels was ascertained by a ROC curve study. The ROC curve showed that AURKB expression had ideal specificity and sensitivity for the diagnosis of all HCC (AUC = 0.977) (Figure 5F). Moreover, the time-dependent ROC curve analysis showed that the 1-year, 3-year, and 5-year AUC values were 0.686, 0.629, and 0.620, respectively (Figure 5G). Together, these results indicate that the up-regulated expression of AURKB is linked to an adverse prognosis in HCC patients.

## 3. Discussion

Due to the complex molecular mechanism of HCC occurrence and development, it is necessary to use bioinformatics methods to perform whole-genome gene expression analysis through microarrays to screen and identify new prognostic markers of HCC. In recent years, many studies have identified new diagnostic and prognostic biomarkers in different types of cancer through bioinformatics methods [35,36,37,38]. Despite the fact that the first Aurora kinase was identified in 1995 and that AURKB has been connected to the development of several malignancies (Appendix A), the links with HCC are not fully understood. AURKB is oncogenic and plays a key role in aneuploidy formation and cancer progression. One of the key characteristics and factors accelerating the development of cancer is aneuploidy, which can be caused by altering either of the mitotic processes. Therefore, we speculate that AURKB plays a crucial role in HCC growth, treatment, and prognosis. 

The phenomenon of AURKB overexpression or increased copy number has been observed in tissues and cells of various cancers, such as lung cancer, gastric cancer, and glioblastoma [39,40]. In this study, we also found similar results in that AURKB was highly expressed in HCC. AURKB was found to be ubiquitously and significantly overexpressed in cancer tissues based on data from the TCGA database and the GEO database. We further validated the expression of AURKB in HCC using the data in the database. The results showed that AURKB mRNA and protein expression levels were significantly increased in HCC. The same results were also found in clinical HCC tissues and HCC cell lines. AURKB is known to be detected in plasma [41], suggesting that AURKB in patients’ serum will be a useful biomarker to detect HCC. 

In previous studies, AURKB has been shown to affect the proliferation and aggressiveness of HCC cells [29]. Therefore, we identified AURKB-associated DEGs by functional enrichment analysis to explore AURKB’s potential biological functions. Our study demonstrated that DEGs associated with the cell cycle by GSEA were enriched in HCC tissues with high expression of AURKB. At the same time, AURKB was also found to be enriched in the resolution of sister chromatid cohesion and mitotic spindle checkpoint pathways. AURKB has been shown to promote the release of sister chromatid adhesions, increasing the risk of chromosome nondisjunction and aneuploidy, and also to maintain normal mitosis and cell cycle checkpoints by activating ATM [42,43]. These results provide supportive evidence that AURKB plays a crucial role in cell mitosis. Therefore, high expression of AURKB promotes HCC development and progression by regulating the cell cycle and mitosis. 

AURKB knockdown was demonstrated in earlier research to suppress HCC cell proliferation in vitro, resulting in cell cycle arrest in the G2/M phase [44]. The use of inhibitors that simultaneously target AURKB is a promising cancer treatment strategy [45,46,47]. A variety of small-molecule inhibitors targeting AURKB that inhibit AURKB self-phosphorylation and histone H3 phosphorylation have been developed. Balacetidine, Hesperadin, SP-96, and others are specific inhibitors of AURKB [48,49,50]. GSK1070916, AT9283, and TAK-901 are pan-AURK inhibitors that have entered clinical trials [51,52,53]. The same results were obtained in our study, in that the protein and mRNA expression levels and proliferation of the HepG2 and PLC5 cells were significantly reduced when AURKB expression was knocked down. Moreover, our findings showed that AURKB knockdown causes G2/M cell cycle arrest in HCC cells. These data indicate that AURKB is a good therapeutic target and prognostic biomarker.

Because AURKB expression levels are significantly elevated in HCC, we evaluated their clinical significance by studying their relationship with multiple clinicopathological characteristics. Our study found that the expression of AURKB was higher in HCC patients, whether male or female, and whether they were elderly or not. Meanwhile, AURKB showed high expression in two different tumor status, tumor-free and with tumor. Regarding the pathological stage, the higher the stage, the higher the AURKB expression. There was a significant difference between the expression levels of AFP and AURKB, and the expression level of AURKB in the AFP > 400 ng/mL group was higher than that in the AFP ≤ 400 ng/mL group. The expression of AURKB was increased in patients of certain races, and AURKB may be used as a biomarker for the detection of liver cancer in different regions. These data indicate that AURKB expression has clinical significance in determining the pathological features of HCC.

It has been reported that AURKB is linked to a poor prognosis in patients with chondrosarcoma [54], neuroblastomas [55], and gastric cancer [56]. Next, we discovered how AURKB expression affected HCC patients’ prognosis. Kaplan–Meier survival analysis revealed that the OS, DSS, and PFI of patients with high AURKB expression were lower than those of patients with low AURKB expression, which is consistent with previously reported findings [57,58]. We also analyzed the relative risks indicated by AURKB in the prognosis of HCC. According to Cox regression analysis, increased expression of AURKB is a risk factor for poor prognosis and death in patients with HCC, as well as an independent predictor of prognosis. Additionally, we examined how AURKB expression affected HCC diagnosis. AURKB expression predicted the prognosis of HCC patients with an AUC value of 0.977. Furthermore, the AUC values for the projected survival after one, three, and five years were all higher than 0.6. These findings indicate that AURKB overexpression has good diagnostic and prognostic potential in HCC. 

However, there are some limitations to our study. Our findings are based on RNA sequencing data from HCC tissues in the TCGA database and the GEO database. Firstly, we would like to point out that we have not yet studied the function and effect of AURKB at the molecular level, such as the activity of the AURKB downstream signaling pathway. Secondly, the clinical sample size collected in this study was relatively small and came from a single center, which may lead to less accurate results. Therefore, our next step will be to investigate the mechanism of AURKB in HCC through cell and animal experiments and even through collecting clinical specimens from multiple centers.

## 4. Materials and Methods

### 4.1. Gene Expression Analysis of AURKB 

To analyze AURKB expression in HCC, RNA sequencing data of 33 human cancers, including HCC, and corresponding normal tissues were extracted from the TCGA database, and the GSE84402 dataset was downloaded from the GEO database [59]. The “DESeq2” (v1.26.0) R package was used to analyze the DEGs between the two groups, and a fold change absolute value of >2 and *p* value of <0.001 were used as threshold parameters for analysis [60]. The R package “ggplot2” (v3.3.6) was used to visualize the results and draw the volcano map.

### 4.2. Functional Enrichment Analysis 

The DEGs associated with AURKB in HCC tissues were converted from entrezIDs into gene symbols using the “org.Hs.eg.db” (v3.10.0) R package. The “ClusterProfiler” (v4.2.1) R package and the reference gene sets obtained from the MgDB file (c2.cp.all.v2022.1.Hs.symbols.gmt [61]) were used for GSEA of the DEGs [62]. 

### 4.3. Cell Lines and Cell Culture 

Four human HCC cell lines (HepG2, PLC5, Huh7, and AD38) and two modified normal liver cell lines (MIHA and LO2) were purchased from the American Type Culture Collection (Manassas, VR, USA). The cells were cultured in Dulbecco’s modified Eagle’s medium (DMEM, Sigma, MO, USA) supplemented with 10% fetal bovine serum (FBS, Lonsera, Uruguay, South America) and 1% penicillin–streptomycin (PS, Cytiva, Washington, DC, USA). All cells were cultured in a cell incubator at 37 °C and 5% CO_2_.

### 4.4. Short Interfering RNA Transfection

To decrease the expression of AURKB in cells, 3 different types of specific small interfering RNAs (siRNAs, 200 pmol) for AURKB knockdown were transfected into HepG2 and PLC5 cells (seeded in 6-well plates) with 5 μL of Lipo8000 transfection reagent (Beyotime, Shanghai, China) and 125 μL of Opti-MEM (Gibco, Grand Island, NY, USA) according to the manufacturer′s protocol, i.e., si-2, si-3, and siNC were used as negative controls. The sequences of the siRNAs were as follows: si-2 sense, GGCACUUCACAAUUGAUGATT; antisense, UCAUCAAUUGUGAAGUGCCTT; si-3 sense, GGCGCAUGCACAAUGAGAATT; antisense, UUCUCAUUGUGCAUGCGCCTT.

### 4.5. Quantitative Real-Time PCR Analysis

Total RNA was extracted from liver tissues and cells using TRNzol Universal Reagent (TIANGEN BIOTECH, Beijing, China). The RNA was used to synthesize cDNA with a PrimeScript™ RT reagent Kit with gDNA Eraser (TaKaRa, Dalian, China). cDNA was used for quantitative real-time PCR (qRT–PCR) with the SYBRPRIME qPCR Kit (Fast HS, BIOGROUND, Chongqing, China). The following primers were used: AURKB forward, 5′-CAGAAGAGCTGCACATTTGACG-3′; AURKB reverse, 5′-CCTTGAGCCCTAAGAGCAGATTT-3′; β-actin forward, 5′-CCTTCCTGGGCATGGAGTC-3′; β-actin reverse, 5′-TGATCTTCATTGTGCTGGGTG-3′.

### 4.6. Western Blotting

After 48 h of cell transfection using Lipo8000™ Transfection Reagent (Beyotime), cells were collected in order to extract proteins from them using cell lysates (RIPA, Sangon Biotech, Shanghai, China; Cocktail, MCE, Shanghai, China). The concentration of protein in the total cell lysates was determined using the Enhanced BCA Protein Assay Kit (Beyotime). The protein was subjected to immunoblotting as described previously using the 12.5% Non-Closure SDS-PAGE Color Preparation kit (Sangon Biotech). The antibodies used were anti-Aurora B antibody (Huabio, Hangzhou, China 1:1000 for WB), anti-glyceraldehyde-3-phosphate dehydrogenase (GAPDH, Zenbio, Chengdu, China, 1:5000 for WB), and HRP-conjugated anti-mouse (Huabio, 1:4000 for WB) or anti-rabbit immunoglobulin antibodies (Huabio, 1:50000 for WB). The expression of proteins was monitored using Immobilon Western Chemiluminescent HRP Substrate (Millipore, Burlington, MA, USA) reagent and visualized with an ECL detection system (Thermo Scientific, Waltham, MA, USA).

### 4.7. Cell Counting Kit-8 (CCK-8) Test

Cells treated with siRNA were placed in 96-well plates at 5 × 10^3^ cells per well. After transfection for 0, 24, 48, or 72 h, the medium was added with 10 μL of CCK-8 (Beyotime) and incubated for 1 h. The optical density was determined on a microplate reader with a 450 nm filter.

### 4.8. Cell Cycle Analysis

At 48 h after transfection of siRNA, cells were collected by centrifugation at 1000× *g* for 3–5 min, washed twice with cold PBS, and fixed with precooled 70% ethanol at 4 °C for 24 h. Afterwards, the cells were resuspended in PBS. Cells were stained using cell cycle and apoptosis detection kits (Beyotime). The cell cycle distribution was determined using flow cytometry (CytoFLEX, Beckman Coulter, Brea, CA, USA). Each assay was performed in triplicate, and the experiment was repeated at least three times.

### 4.9. Statistical Analysis

Clinicopathological data for HCC patients were obtained from the TCGA database and a previously published study on HCC patients [63]. The differences between groups of the clinicopathological parameters of patients were analyzed by using the Shapiro–Wilk normality test, Kruskal–Wallis test, and Dunn’s multiple hypothesis test. The “survival” package (3.3.1) was used to test the proportional risk hypothesis and to fit the survival regression curve. The significant factors for overall survival were identified by univariate Cox regression analysis and introduced into the multivariate Cox regression model. ROC analysis of the data was performed using the pROC and timeROC packages to determine the diagnostic significance of AURKB expression for HCC. All statistical analyses were performed through the R package stats (v4.2.1), and statistical significance was acknowledged in case of *p* < 0.05.

## 5. Conclusions

In summary, our study revealed that AURKB expression is associated with the development of HCC. The results showed that AURKB was overexpressed in HCC samples. AURKB could regulate HCC progression through the cell cycle. High expression of AURKB was associated with a shorter survival time; thus, AURKB can not only be used as an independent predictor, but also provide a reliable basis for the diagnosis of HCC. Therefore, AURKB is a promising diagnostic and prognostic biomarker and a potential therapeutic target for HCC.

## Figures and Tables

**Figure 1 ijms-25-02199-f001:**
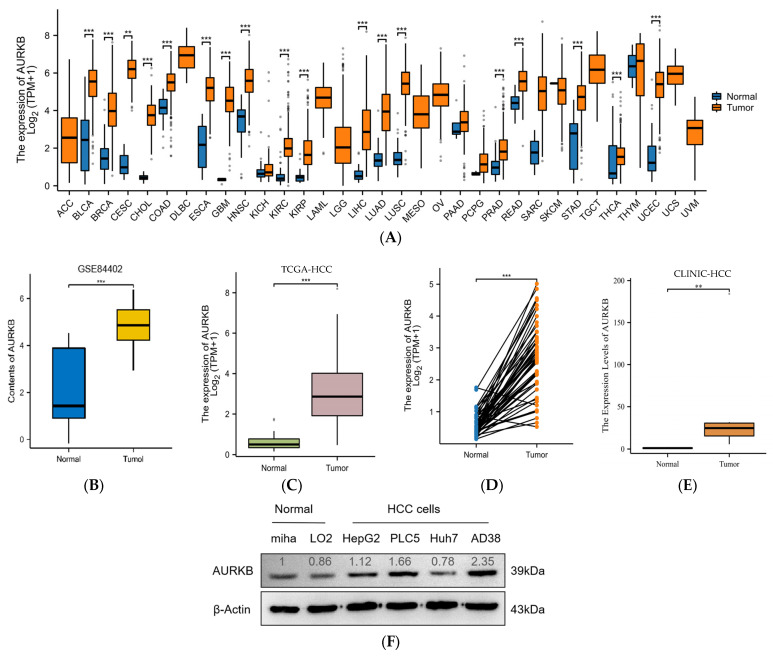
AURKB expression is up-regulated in HCC. (**A**) Comparison of the expression of AURKB in tumors and adjacent normal tissues across 33 cancers including HCC based on TCGA database. *** *p* < 0.001. (**B**,**C**) AURKB expression levels in normal and tumor tissues of the liver were assessed in the GSE84402 dataset (**B**) and the TCGA database (**C**). *** *p* < 0.001. (**D**,**E**) AURKB mRNA shown in paired HCC and normal tissues in the same patients were assessed in (**D**) TCGA database and (**E**) clinical HCC samples. ** *p* < 0.01; ***, *p* < 0.001. (**F**) Western blotting of AURKB expression in normal and tumor liver cell lines. (**G**) Immunohistochemistry of AURKB expression in normal and tumors tissues obtained from the Human Protein Atlas database.

**Figure 2 ijms-25-02199-f002:**
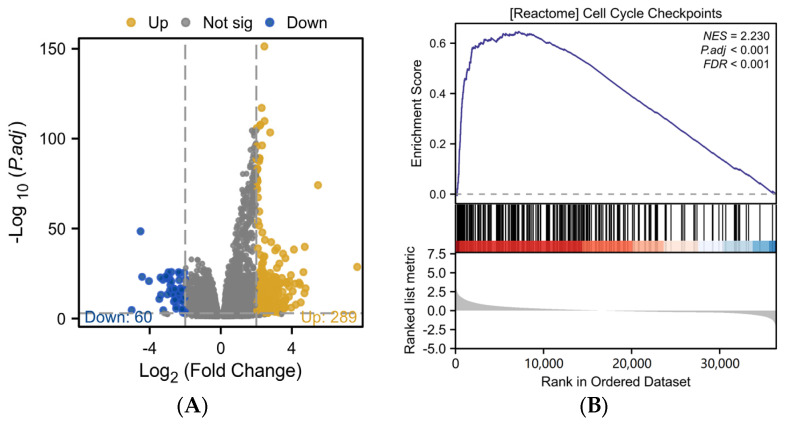
Differential expression analysis and functional enrichment analysis of AURKB in HCC. (**A**) Volcano plot analysis of DEGs associated with AURKB. (**B**–**D**) GSEA enrichment of DEGs associated with AURKB revealed that there were significant enrichment up-regulated pathways: (**B**) CELL_CYCLE_CHECKPOINTS, (**C**) RESOLUTION_OF_SISTER_CHROMATID_COHESION, (**D**) MITOTIC_SPINDLE_CHECKPOINT.

**Figure 3 ijms-25-02199-f003:**
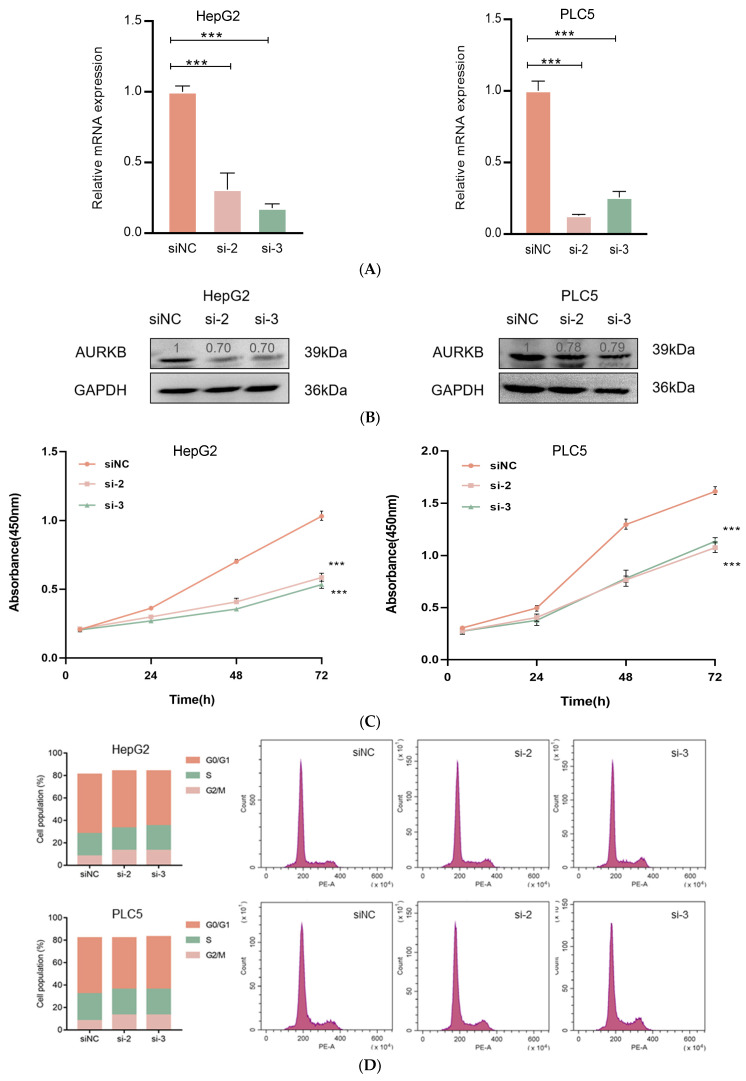
AURKB knockdown inhibits HCC cell proliferation. (**A**) qRT-PCR was used to analyze the expression of AURKB mRNA. ***, *p* < 0.001. (**B**) Western blot was used to analyze AURKB protein expression. (**C**) CCK-8 was used to analyze the effects of siRNA knockout of AURKB on cell proliferation. ***, *p* < 0.001. (**D**) Flow cytometry was used to analyze the cell cycle distribution after siRNA (siNC, si-2, si-3) knockout of AURKB.

**Figure 4 ijms-25-02199-f004:**
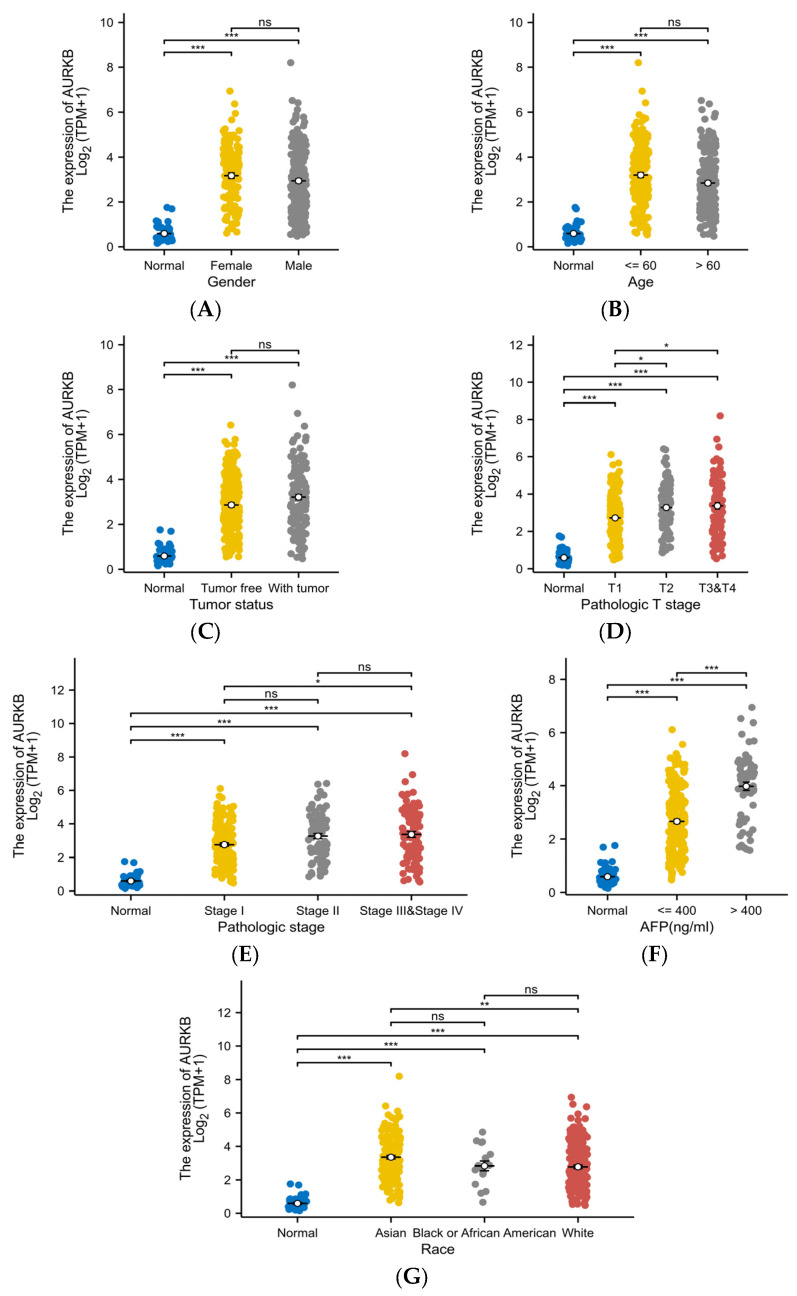
Correlation of AURKB expression with different clinicopathological characteristics. The correlation analysis under different clinical subgroups: (**A**) gender, (**B**) age, (**C**) tumor status, (**D**) T stage, (**E**) pathologic stage, (**F**) AFP level, and (**G**) race. * *p* < 0.05; ** *p* < 0.01; *** *p* < 0.001; ns, not significant.

**Figure 5 ijms-25-02199-f005:**
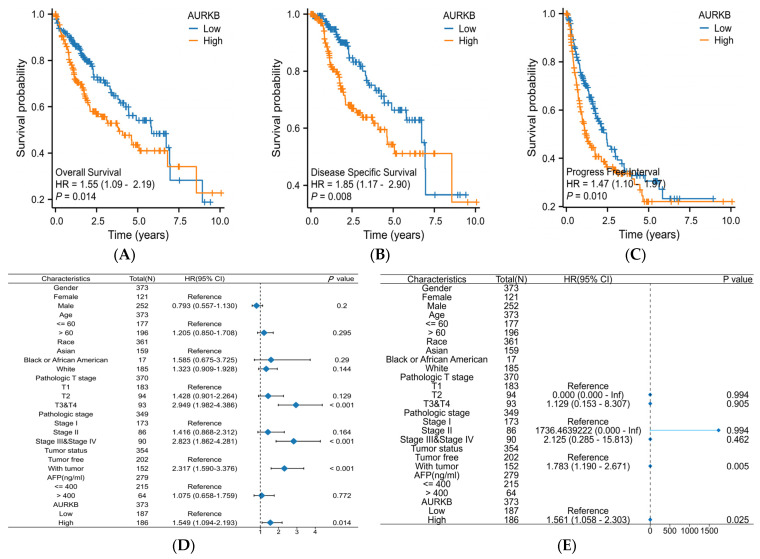
Evaluation of the prognostic and diagnostic value of AURKB in HCC. (**A**–**C**) Kaplan–Meier survival analysis based on AURKB expression in terms of (**A**) overall survival, (**B**) disease-specific survival, and (**C**) progression-free interval. *p* < 0.05. (**D**,**E**) Forest plot showing the hazard ratio (HR) with 95% confidence interval (CI) for HCC patients based on (**D**) univariate Cox analysis and (**E**) multivariate Cox analysis. (**F**) ROC curve analysis of AURKB in HCC. (**G**) Time-dependent ROC curves of AURKB in HCC.

**Table 1 ijms-25-02199-t001:** GSEA of the signaling pathways based on the AURKB-associated DEGs.

Gene Set Name	NES	*p*.adj	q Value
REACTOME_RESOLUTION_OF_SISTER_CHROMATID_COHESION	2.271689866	4.34483 × 10^−9^	3.26134 × 10^−9^
REACTOME_MITOTIC_PROMETAPHASE	2.234457337	4.34483 × 10^−9^	3.26134 × 10^−9^
REACTOME_CELL_CYCLE_CHECKPOINTS	2.230407175	4.34483 × 10^−9^	3.26134 × 10^−9^
REACTOME_MITOTIC_SPINDLE_CHECKPOINT	2.188956127	4.34483 × 10^−9^	3.26134 × 10^−9^
REACTOME_MITOTIC_METAPHASE_AND_ANAPHASE	2.159328603	4.34483 × 10^−9^	3.26134 × 10^−9^
REACTOME_RHO_GTPASES_ACTIVATE_FORMINS	2.157610171	4.34483 × 10^−9^	3.26134 × 10^−9^
REACTOME_SEPARATION_OF_SISTER_CHROMATIDS	2.140987180	4.34483 × 10^−9^	3.26134 × 10^−9^
REACTOME_M_PHASE	2.062197669	4.34483 × 10^−9^	3.26134 × 10^−9^
REACTOME_RHO_GTPASE_EFFECTORS	1.930962445	4.34483 × 10^−9^	3.26134 × 10^−9^
PID_AURORA_B_PATHWAY	2.166339108	9.80933 × 10^−7^	7.36315 × 10^−7^
WP_COHESIN_COMPLEX_CORNELIA_DE_LANGE_SYNDROME	2.144033450	1.50116 × 10^−5^	1.12681 × 10^−5^
REACTOME_APC_C_MEDIATED_DEGRADATION_OF_CELL_CYCLE_PROTEINS	1.970492568	1.88795 × 10^−5^	1.41714 × 10^−5^
REACTOME_SUMOYLATION_OF_DNA_REPLICATION_PROTEINS	1.906826736	0.000492123	0.000369401
REACTOME_TRANSCRIPTIONAL_REGULATION_BY_TP53	1.507967615	0.00069652	0.000522827
REACTOME_REGULATION_OF_TP53_ACTIVITY	1.617215716	0.001740339	0.001306344
PID_FOXM1_PATHWAY	1.875676222	0.001867753	0.001401985

NES, normalized enrichment score. Gene sets with |NES| > 1, *p*.adj < 0.05, and q value of < 0.25 were regarded as significantly enriched.

**Table 2 ijms-25-02199-t002:** The relationship between the clinicopathological characteristics and AURKB expression levels in HCC.

Characteristics	Expression Level of AURKB				*p* Value
	Low AURKB		High AURKB		
	Total (187)	Percentages (%)	Total (187)	Percentages (%)	
Gender, n (%)					0.011
Female	49	40.5%	72	59.5%	
Male	138	54.5%	115	45.5%	
Age, n (%)					0.070
≤60	80	45.2%	97	54.8%	
>60	107	54.6%	89	45.4%	
Race, n (%)					0.006
Asian	65	40.6%	95	59.4%	
Black or African American	9	52.9%	8	47.1%	
White	107	57.8%	78	42.2%	
Pathologic T stage, n (%)					0.012
T1	105	58.5%	78	41.5%	
T2	39	41.1%	56	58.9%	
T3 or T4	40	43.0%	53	57.0%	
Pathologic stage, n (%)					0.028
Stage I	98	56.6%	75	43.4%	
Stage II	37	42.5%	50	57.5%	
Stage III or Stage IV	38	42.2%	52	57.8%	
Tumor status, n (%)					0.092
Tumor-free	108	53.5%	94	46.5%	
With tumor	68	44.4%	85	55.6%	
AFP (ng/mL), n (%)					<0.001
≤400	128	59.5%	87	40.5%	
>400	16	24.6%	49	75.4%	

**Table 3 ijms-25-02199-t003:** Univariate and multivariate Cox regression analysis of HCC patients.

Characteristics	Total (N)	Univariate Cox		Multivariate Cox	
		HR (95% CI)	*p*	HR (95% CI)	*p*
Gender	373				
Female	121	1			
Male	252	0.793 (0.557–1.130)	0.200		
Age	373				
≤60	177	1			
>60	196	1.205 (0.850–1.708)	0.295		
Race	361				
Asian	159	1			
Black or African American	17	1.585 (0.675–3.725)	0.290		
White	185	1.323 (0.909–1.928)	0.144		
Pathologic T stage	370				
T1	183	1		1	
T2	94	1.428 (0.901–2.264)	0.129	0.000 (0.000–Inf)	0.994
T3 or T4	93	2.949 (1.982–4.386)	<0.001	1.129 (0.153–8.307)	0.905
Pathologic stage	349				
Stage I	173	1		1	
Stage II	86	1.416 (0.868–2.312)	0.164	1736463.9222 (0.000–Inf)	0.994
Stage III or Stage IV	90	2.823 (1.862–4.281)	<0.001	2.125 (0.285–15.813)	0.462
Tumor status	354				
Tumor-free	202	1		1	
With tumor	152	2.317 (1.590–3.376)	<0.001	1.783 (1.190–2.671)	0.005
AFP (ng/mL)	279				
≤400	215	1			
>400	64	1.075 (0.658–1.759)	0.772		
AURKB	373				
Low	187	1		1	
High	186	1.549 (1.094–2.193)	0.014	1.561 (1.058–2.303)	0.025

## Data Availability

The databases used in this study are publicly available and can be obtained from the following: TCGA: https://portal.gdc.cancer.gov/, accessed on 15 May 2023; GEO: https://www.ncbi.nlm.nih.gov/geo/, accessed on 20 July 2023; TCGA-LIHC: https://portal.gdc.cancer.gov/, accessed on 13 August 2023; GSE84402: https://www.ncbi.nlm.nih.gov/geo/, accessed on 26 August 2023.

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
