# Peer review of "Overexpression of Aurora Kinase B Is Correlated with Diagnosis and Poor Prognosis in Hepatocellular Carcinoma"

_ijms, 2024, doi:10.3390/ijms25042199_

Round 1

Reviewer 1 Report

Comments and Suggestions for Authors

Dear authors,

I think that the manuscript presents interesting, important data in relation to HCC. It has some weak sides, which are mentioned at the end of the paper. For me, lack of molecular mechanisms of such an overexpression is quite disappointing me, but it's a chance to expand studies in the future.

Introduction section should be expanded. More information about deregulation of AURKB in cancer should be mentioned. Discussion section is also too general. More data should be presented about specific inhibitors, which are tested in pre-clinical and clinical trials. More references should be cited in this paper.

Comments on the Quality of English Language

Minor spelling should be checked and corrected

Author Response

Thank you for your letter and the constructive comments on this article in your busy schedule. All of us authors have carefully read the comments that you have given us, and have discussed and revised each of these issues, the detailed corrections are listed below. In addition, we have resubmitted a new maniscript in the revised state, with the revisions highlighted in red. If there are any incorrect answers or questions in the manuscript, please do not hesitate to let us know.

  1. I think that the manuscript presents interesting, important data in relation to HCC. It has some weak sides, which are mentioned at the end of the paper. For me, lack of molecular mechanisms of such an overexpression is quite disappointing me, but it's a chance to expand studies in the future.

Thank you very much for the valuable comments from the reviewer. We strongly agree with your proposal on the molecular mechanism of the lack of AURKB overexpression, which is exactly our next research plan.

  1. Introduction section should be expanded. More information about deregulation of AURKB in cancer should be mentioned. Discussion section is also too general. More data should be presented about specific inhibitors, which are tested in pre-clinical and clinical trials. More references should be cited in this paper.

1. As you mentioned that the introduction section needs to be expanded, we have supplemented its content, and the information you mentioned that you want to increase the AURKB disorder has also been added, see for detailsP2, line 62-74.

2. We really lack in-depth analysis in the discussion section, we have made corresponding changes, and the content you mentioned about specific inhibitors has also been added, see for details P13,line 220-226.

3. As suggested by the reviewer, we have added more references, and by looking for more study further verify the reliability of the results of our study

4. We tried our best to improve the manuscript and made some changes to the manuscript. These changes will not influence the content and framework of the paper. And here we did not list the changes but marked in red in the revised paper. We apprecite for reviewer warm work earnestly and hope that the correction will meet with approval.

Reviewer 2 Report

Comments and Suggestions for Authors

1.       Although the study was conducted based on analyses of several data base, there are not explanation of data contents. Especially, they demonstrated a lot of clinicopathological analyses but nothing to explain them, although they are critical results in their study.

2.       Four samples of IHC are too small to say anything.

3.       In Figure 1C and D, both are results of TCGA analyses, aren’t they? Repeated descriptions?

4.       In the manuscript, authors describe Figure 1A, 1B, and so on, but (a), (b) in Figures. It should be same.

5.       Although authors claim that AURKB expression is higher in cancer cell lines than those in normal cell lines in Figure 1F, it looks wrong interpretation. HepG2 has the lowest expression, doesn’t it?

6.       Although authors have done gene enrichment analysis of DEGs and demonstrated correlation of AURKB and just cell cycle relating genes, they should show all results of enriched pathway. It is important to compare known relating pathway and unknown pathways.

7.       Although authors have done knock-down experiments, effects of gene knock-down are not acceptable levels. They can’t say anything from those results.

8.       P7, line174, there are different font.

9.       Although authors discuss about elevated expression of AURKB in the serum of HCC patients, there are no description of patient’s serum in the manuscript.

10.    There are many repeated descriptions.

Author Response

Thank you for your letter and the constructive comments on this article in your busy schedule. All of us authors have carefully read the comments that you have given us, and have discussed and revised each of these issues, the detailed corrections are listed below. In addition, we have resubmitted a new maniscript in the revised state, with the revisions highlighted in red. If there are any incorrect answers or questions in the manuscript, please do not hesitate to let us know.

  1. Although the study was conducted based on analyses of several data base, there are not explanation of data contents. Especially, they demonstrated a lot of clinicopathological analyses but nothing to explain them, although they are critical results in their study.

We did explain too little about the data content of the clinicopathological analysis. As suggested by the reviewer, we explained and discussed the results in the results and discussion section, see for details P7, line 149-152, and P13, line 233-241.

  1. Four samples of IHC are too small to say anything

Thank you for reviewing experts consider so stringent, we are through public database retrieval (https://www.proteinatlas.org/) for this part. In the HPA website, a total of 21 cases of immunohistochemical results of liver cancer were included, including 12 cases of hepatocellular carcinoma, and the staining intensity was strong in 2 cases, weak in 4 cases, and moderate in 2 cases. We have selected the typical results to show in Figure 1G.

  1. In Figure 1C and D, both are results of TCGA analyses, aren’t they? Repeated descriptions?

As you said, the data used in Figure 1C and Figure 1D are from TCGA, but the samples used in Figure 1C for the analysis of AURKB expression level are unpaired samples, while the samples used in Figure 1D are paired samples. Our previous description may indeed be ambiguous, and the details have been revised in P2, line 94-98.

  1. In the manuscript, authors describe Figure 1A, 1B, and so on, but (a), (b) in Figures. It should be same.

As suggested by the reviewer, in our submitted manuscript, Figures 1A, 1B, and so on in our description and A/B in Figure are indeed consistent, but may have changed after uploads due to differences in format. According to your comments, we have revised all the contents involved in the full text.

  1. Although authors claim that AURKB expression is higher in cancer cell lines than those in normal cell lines in Figure 1F, it looks wrong interpretation. HepG2 has the lowest expression, doesn’t it?

Thank the reviewer for finding this problem, this is indeed a major mistake on our part, and we are sorry for our castedness. We checked the original records and found that there were mistakes in the marking of the lanes of HepG2 and Huh7, and the expression level of Huh7 should be the lowest. We again checked our manuscript word for word, word for picture.

  1. Although authors have done gene enrichment analysis of DEGs and demonstrated correlation of AURKB and just cell cycle relating genes, they should show all results of enriched pathway. It is important to compare known relating pathway and unknown pathways.

As suggested by the reviewer, we have only shown one of the many pathways enriched by AURKB, Now all the enriched pathways have been plotted Table 1, and 3 pathways have been selected to be plotted Figure 2B, 2C, 2D, as described in P4, line 116-123, and P13, line 210-218.

  1. Although authors have done knock-down experiments, effects of gene knock-down are not acceptable levels. They can’t say anything from those results.

The WB results of the knockdown experiments are indeed less obvious, and we have redone the experiments and updated Figure 3B, simultaneously We are still continuing to optimize the conditions and will update them again after good results.

  1. P7, line174, there are different font.

The different font in line 174 of P7 may also be due to the new problem after the format change. We have revised and checked the whole article to avoid the same error again.

  1. Although authors discuss about elevated expression of AURKB in the serum of HCC patients, there are no description of patient’s serum in the manuscript.

Our description of this is indeed less rigorous, and we expect it to be also well detected in the patient serum, but the discussion here directly describing “AURKB expression in the serum of HCC patients” is indeed problematic. Thanks to the reviewers for this question, which gives us a new thinking, and we have revised it, see for details P13, line 205-206.

  1. There are many repeated descriptions.

We again to check and modify the article as a whole, including where it was repeatedly expressed.

We tried our best to improve the manuscript and made some changes to the manuscript. These changes will not influence the content and framework of the paper. And here we did not list the changes but marked in red in the revised paper. We apprecite for reviewer warm work earnestly and hope that the correction will meet with approval.

Round 2

Reviewer 1 Report

Comments and Suggestions for Authors

Authors improved their work, I accept in present form. 

Author Response

We would like to thank you for your professional review work, constructive comments, and valuable suggestions on our manuscript.

Reviewer 2 Report

Comments and Suggestions for Authors

Thank you for your responses to my comments. I find some improvements but still not enough to be published. My comments are below in bold.

Thank you for your letter and the constructive comments on this article in your busy schedule. All of us authors have carefully read the comments that you have given us, and have discussed and revised each of these issues, the detailed corrections are listed below. In addition, we have resubmitted a new maniscript in the revised state, with the revisions highlighted in red. If there are any incorrect answers or questions in the manuscript, please do not hesitate to let us know.

  1. Although the study was conducted based on analyses of several data base, there are not explanation of data contents. Especially, they demonstrated a lot of clinicopathological analyses but nothing to explain them, although they are critical results in their study.

We did explain too little about the data content of the clinicopathological analysis. As suggested by the reviewer, we explained and discussed the results in the results and discussion section, see for details P7, line 149-152, and P13, line 233-241.

I can’t find any corrections in line 147-150 on P7 of revised MS. It looks identical to those in original MS.

  1. Four samples of IHC are too small to say anything

Thank you for reviewing experts consider so stringent, we are through public database retrieval (https://www.proteinatlas.org/) for this part. In the HPA website, a total of 21 cases of immunohistochemical results of liver cancer were included, including 12 cases of hepatocellular carcinoma, and the staining intensity was strong in 2 cases, weak in 4 cases, and moderate in 2 cases. We have selected the typical results to show in Figure 1G.

 It is still too few. Explanations of sample information is not enough and sample numbers you indicated above is different from you described in MS. 18 cancer tissues in MS. It should be described details correctly in MS.

Although there is information of just 4 samples in Supplementary Table S1, it should be shown those of all samples.

The description about clinico-pathological analysis is still tough to understand. What does it mean, the AURKB expression level was significantly correlated with gender, age, tumor status, T stage, pathologic stage AFP level, and race? Does it mean, the AURKB expression level was significantly higher in female, in patients <= 60 years old, in patients with advanced stage, patients with higher AFP level, and in Asian? But I don’t find any differences between gender, age, tumor status.

I also propose to change indications of % in Table 2. What do you want to show in this table? For example, it is better to indicate that female: low AURKB 49 (40.5%) and high AURKB 72 (59.5%), male: 138 (54.5%) and 115 (45.5%), although it is showing that. female: low AURKB 49 (13.1%) and high AURKB 72 (19.3%), male: 138 (36.9%) and 115 (30.7%) in present Table 2. It makes easier to understand that positivity of high-AURKB is higher in female.

  1. In Figure 1C and D, both are results of TCGA analyses, aren’t they? Repeated descriptions?

As you said, the data used in Figure 1C and Figure 1D are from TCGA, but the samples used in Figure 1C for the analysis of AURKB expression level are unpaired samples, while the samples used in Figure 1D are paired samples. Our previous description may indeed be ambiguous, and the details have been revised in P2, line 94-98.

 OK, it is improved.

  1. In the manuscript, authors describe Figure 1A, 1B, and so on, but (a), (b) in Figures. It should be same.

As suggested by the reviewer, in our submitted manuscript, Figures 1A, 1B, and so on in our description and A/B in Figure are indeed consistent, but may have changed after uploads due to differences in format. According to your comments, we have revised all the contents involved in the full text.

 OK, it is improved.

  1. Although authors claim that AURKB expression is higher in cancer cell lines than those in normal cell lines in Figure 1F, it looks wrong interpretation. HepG2 has the lowest expression, doesn’t it?

Thank the reviewer for finding this problem, this is indeed a major mistake on our part, and we are sorry for our castedness. We checked the original records and found that there were mistakes in the marking of the lanes of HepG2 and Huh7, and the expression level of Huh7 should be the lowest. We again checked our manuscript word for word, word for picture.

 This is critical mistake. I find that it is still wrong data in supplemented original images. How can we trust you? And I have noticed that you have done this experiment just once, since just one image was submitted. How did you calculate significance then, although you described as “significantly higher levels of AURKB expression” in MS?

You described as “four human HCC cell lines (Huh7, PLC5, HepG2, and AD38)” in MS but “HepG2, PLC5, Huh7, and AD38” in Figure 1F. It is better to indicate names as same order to avoid any mistakes.

  1. Although authors have done gene enrichment analysis of DEGs and demonstrated correlation of AURKB and just cell cycle relating genes, they should show all results of enriched pathway. It is important to compare known relating pathway and unknown pathways.

As suggested by the reviewer, we have only shown one of the many pathways enriched by AURKB, Now all the enriched pathways have been plotted Table 1, and 3 pathways have been selected to be plotted Figure 2B, 2C, 2D, as described in P4, line 116-123, and P13, line 210-218.

 OK, it is improved.

  1. Although authors have done knock-down experiments, effects of gene knock-down are not acceptable levels. They can’t say anything from those results.

The WB results of the knockdown experiments are indeed less obvious, and we have redone the experiments and updated Figure 3B, simultaneously We are still continuing to optimize the conditions and will update them again after good results.

 It seems to be not improved yet. If you want to show in this paper, you have to improve in this time.

  1. P7, line174, there are different font.

The different font in line 174 of P7 may also be due to the new problem after the format change. We have revised and checked the whole article to avoid the same error again.

 It seems to be not improved yet. It is still existing.

  1. Although authors discuss about elevated expression of AURKB in the serum of HCC patients, there are no description of patient’s serum in the manuscript.

Our description of this is indeed less rigorous, and we expect it to be also well detected in the patient serum, but the discussion here directly describing “AURKB expression in the serum of HCC patients” is indeed problematic. Thanks to the reviewers for this question, which gives us a new thinking, and we have revised it, see for details P13, line 205-206.

 It is still confusing and misleading. Do you mean that “AURKB known to be detected in plasma [references], suggesting that AURKB in patients’ serum will be a useful biomarker to detect HCC.”? Please revise correctly.

  1. There are many repeated descriptions.

We again to check and modify the article as a whole, including where it was repeatedly expressed.

 OK, but please check it again.

Author Response

Thank you for your letter and the constructive comments on this article in your busy schedule. All of us authors have carefully read the comments that you have given us, and have discussed and revised each of these issues, the detailed corrections are listed below. In addition, we have resubmitted a new maniscript in the revised state, with the revisions highlighted in red. At the first rework the reviewer's comments and our response are what follows the serial number (1), and at the second rework the reviewer's comments and our response are what follows the serial number (2). If there are any incorrect answers or questions in the manuscript, please do not hesitate to let us know.

  1. (1)Although the study was conducted based on analyses of several data base, there are not explanation of data contents. Especially, they demonstrated a lot of clinicopathological analyses but nothing to explain them, although they are critical results in their study.

We did explain too little about the data content of the clinicopathological analysis. As suggested by the reviewer, we explained and discussed the results in the results and discussion section, see for details P7, line 149-152, and P13, line 233-241.

(2)I can’t find any corrections in line 147-150 on P7 of revised MS. It looks identical to those in original MS.

Thanks to the reviewer for raising this question again. In the first revision, there were some deviations in our understanding of this suggestion, which led to the omission of this revision.We looked at the manuscript and there was really less description of Table2.We think this point pointed out by the reviewers is indeed very good.As suggested by the reviewer, our description of the results has been revised, as can be seen in lines 148-156 on P7.

  1. (1)Four samples of IHC are too small to say anything

Thank you for reviewing experts consider so stringent, we are through public database retrieval (https://www.proteinatlas.org/) for this part. In the HPA website, a total of 21 cases of immunohistochemical results of liver cancer were included, including 12 cases of hepatocellular carcinoma, and the staining intensity was strong in 2 cases, weak in 4 cases, and moderate in 2 cases. We have selected the typical results to show in Figure 1G.

 ï¼ˆ2)It is still too few. Explanations of sample information is not enough and sample numbers you indicated above is different from you described in MS. 18 cancer tissues in MS. It should be described details correctly in MS.

Although there is information of just 4 samples in Supplementary Table S1, it should be shown those of all samples.

The description about clinico-pathological analysis is still tough to understand. What does it mean, the AURKB expression level was significantly correlated with gender, age, tumor status, T stage, pathologic stage AFP level, and race? Does it mean, the AURKB expression level was significantly higher in female, in patients <= 60 years old, in patients with advanced stage, patients with higher AFP level, and in Asian? But I don’t find any differences between gender, age, tumor status.

I also propose to change indications of % in Table 2. What do you want to show in this table? For example, it is better to indicate that female: low AURKB 49 (40.5%) and high AURKB 72 (59.5%), male: 138 (54.5%) and 115 (45.5%), although it is showing that. female: low AURKB 49 (13.1%) and high AURKB 72 (19.3%), male: 138 (36.9%) and 115 (30.7%) in present Table 2. It makes easier to understand that positivity of high-AURKB is higher in female.

We will explain several problems involved in the immunohistochemical results in Figure1G.

Firstly, the reviewer pointed out that the number of samples mentioned in the immunohistochemical results section of our manuscript was different from the 18 cancer tissues described in Figure1A of the manuscript.This is because the cancer mainly studied in this study is HCC, so in Figure1G we only show the immunohistochemical results of HCC related to our study.This is because we want to further assist in verifying the high expression of AURKB in human HCC tissues through immunohistochemical results.

Secondly, we found all the liver tissue immunohistochemical results related to AURKB from The Human Protein Atlas database( https://www.proteinatlas.org/ENSG00000178999-AURKB/pathology/liver+cancer#Location), with a total of 21images. The 21 immunohistochemical results diagrams involved 15 patients in total, and the specific information of all patients has been shown in SupplementaryTableS1. Three of the patients had normal liver tissue and 12 had liver cancer tissue. In the database related AURKB there are two kinds of liver cancer, HCC (Hepatocellularcarcinoma) and CHOL (Cholangiocarinoma). The cancer tissue shown in Figure1G is HCC, which is also because this study discussed the relationship between AURKB's diagnosis of HCC and poor prognosis.

Then, thanks for the reviewer's explanation on this issue. Your description gives us a new view on the interpretation of this result. Our description of this part of the manuscript does not do a good job of explaining the results clearly. We have revised the results of this section and see for details in lines 157-164 on P7.

Finally, As demonstrated in Table 2, for different clinicopathological characteristics, we divided all 374 HCC samples in the TCGA database into two groups of 187 HCC patients with low AURKB expression and 187 HCC patients with high AURKB expression by target gene expression levels. The percentage representation in the table means the proportion of the subgroup containing each clinical case feature to the total number of people (374) in the AURKB low and high expression groups in this clinical case feature category. For example, there were 49 cases of low AURKB expression in females in the sex subgroup, accounting for 13.1% of all HCC samples (374). We did this analysis to analyze whether there was a difference between the low and high expression of AURKB in each clinical case feature. For the analysis between its subgroups we show it in Figure 4.

  1. (1)Although authors claim that AURKB expression is higher in cancer cell lines than those in normal cell lines in Figure 1F, it looks wrong interpretation. HepG2 has the lowest expression, doesn’t it?

Thank the reviewer for finding this problem, this is indeed a major mistake on our part, and we are sorry for our castedness. We checked the original records and found that there were mistakes in the marking of the lanes of HepG2 and Huh7, and the expression level of Huh7 should be the lowest. We again checked our manuscript word for word, word for picture

(2)This is critical mistake. I find that it is still wrong data in supplemented original images. How can we trust you? And I have noticed that you have done this experiment just once, since just one image was submitted. How did you calculate significance then, although you described as “significantly higher levels of AURKB expression” in MS?

You described as “four human HCC cell lines (Huh7, PLC5, HepG2, and AD38)” in MS but “HepG2, PLC5, Huh7, and AD38” in Figure 1F. It is better to indicate names as same order to avoid any mistakes.

I am deeply sorry that the error in the experimental mark in the original picture was not corrected. At the time of the first revision we did not submit new original images and original images of repeated experiments again.Now we have fixed all the errors. 

And we did repeat the experiment 3 times when selecting the cell line at the beginning of the experiment, and now we provide all the original images. For the significance analysis problem, we have added the gray value of the strip in Figure 1F. 

Thanks for the reviewer's valuable comments on our description in the manuscript. Indeed, the order of the different cells described in the manuscript did not match the order in Figure 1F. Keeping the order consistent between the two can effectively avoid errors. We have made the changes according to the recommendations of the reviewers. See P3, line 100 for details.

  1. (1)Although authors have done knock-down experiments, effects of gene knock-down are not acceptable levels. They can’t say anything from those results.

The WB results of the knockdown experiments are indeed less obvious, and we have redone the experiments and updated Figure 3B, simultaneously We are still continuing to optimize the conditions and will update them again after good results.

  • (2)It seems to be not improved yet. If you want to show in this paper, you have to improve in this time.

The WB results of the knockout experiments were indeed less obvious, and we re-optimized the experimental conditions and updated Figure 3B.à 

  1. (1)P7, line174, there are different font.

The different font in line 174 of P7 may also be due to the new problem after the format change. We have revised and checked the whole article to avoid the same error again.

à(2) It seems to be not improved yet. It is still existing.

 Thanks to the reviewers for raising this issue again.After careful examination, we found that the font for cell lines in each of the result plots in Figure 3 did change due to formatting issues, and we have now made adjustments to make it consistent with the font in the result plots.

  1. (1)Although authors discuss about elevated expression of AURKB in the serum of HCC patients, there are no description of patient’s serum in the manuscript.

Our description of this is indeed less rigorous, and we expect it to be also well detected in the patient serum, but the discussion here directly describing “AURKB expression in the serum of HCC patients” is indeed problematic. Thanks to the reviewers for this question, which gives us a new thinking, and we have revised it, see for details P13, line 205-206.

à ï¼ˆ2)It is still confusing and misleading. Do you mean that “AURKB known to be detected in plasma [references], suggesting that AURKB in patients’ serum will be a useful biomarker to detect HCC.”? Please revise correctly.

Thanks to the advice of the reviewers, we have expressed the meaning in the manuscript as you suggested.We have made the modification according to your suggestion, for details, see lines 217-218 on P13.

  1. (1)There are many repeated descriptions.

We again to check and modify the article as a whole, including where it was repeatedly expressed.

  • (3)OK, but please check it again.

We again to check and modify the article as a whole, including where it was repeatedly expressed.

We tried our best to improve the manuscript and made some changes to the manuscript. These changes will not influence the content and framework of the paper. And here we did not list the changes but marked in red in the revised paper. We apprecite for reviewer warm work earnestly and hope that the correction will meet with approval.

Round 3

Reviewer 2 Report

Comments and Suggestions for Authors

Revision is still not enough to be published. Scientific quality must be improved.

For different clinicopathological features, 374 HCC patients in the TCGA database were divided into 187 HCC patients with low AURKB expression and 187 HCC patients with high AURKB expression according to target gene expression levels. Among them, in gender, race, clinical T stage and pathological stage, there were differences between low-AURKB expression and high- AURKB expression in HCC patients (p < 0.05). At the same time, there was a significant difference between low AURKB expression and high AURKB expression in HCC patients at AFP level (p < 0.001), but no association in other clinicopathological characteristics.

Explanation is increased but not enough to understand their results. What does it mean that “low AURKB expression” and “high AURKB expression”? How are they classified? What does it mean “according to target gene expression”?

It is still not easy to understand Table 2.

AURKB expression was significantly higher in subgroups with different clinical case characteristics than normal tissues, including gender (Figure 4A), age (Figure 4B), tumor status (Figure 4C), T stage (Figure 4D), pathological stage (Figure 4E), AFP level (Figure 4F), and race (Figure 4G) (p<0.001).

It doesn’t make sense to compare as above. All are just same as comparing between normal and cancer. When there are any differences between gender, age, tumor status, and stages, you can say that. Among pathological stages, AFP levels, and Races, we can see differences of AURKB expression.

Similarly, immunohistochemical results were obtained from the HPA database for a total of 15 patients, including normal liver tissue from 3 patients, liver tissue from 6 patients with HCC, and liver tissue from 6 patients with Cholangiocarinoma (CHOL). The patient data corresponding to the IHC samples are listed (supplemental Table S1). Immunohistochemical results of normal liver tissues of 2 patients and liver tissue from 2 patients with HCC were selected (Figure 1G), and the results showed that AURKB expression in HCC tissue specimens was higher than that in normal tissue specimens.

 It is still tough to understand above. Does “including normal liver tissue from 3 patients, liver tissue from 6 patients with HCC, and liver tissue from 6 patients with Cholangiocarinoma (CHOL).” mean that “including 3 normal liver tissues from 3 HCC patients, 6 HCC tumor tissues from 6 HCC patients, and 6 CHOL tissues from 6 patients with Cholangiocarcinoma.”? Does it mean you have done IHC analysis of 15 tissue samples? I this case, why don’t you show the summary of IHC analysis in table instead of those of just 4 samples?

 In general, we show positivity of IHC analysis instead of expression levels.

 I still find wrong data in supplemented original images. How can we trust you? 

 Western blotting of knock-down experiment seems to be little bit improved. How many was it repeated? It seems not to be changed growth curve, although western blotting was changed.

Author Response

Dear reviewer, 

Thank you for your email. We sincerely thank the editor and all reviewers for their valuable eedback that we have used to improve the quality of our manuscript (ijms-2735583). Comments from reviewers are shown below in bold type and specific concerns have been numbered. Our response is given in normal font and changes/additions to the manuscript are given in the red text. The changed file has been upload attachments. If there are any incorrect answers or questions in the manuscript, please do not hesitate to let us know.

  1. Explanation is increased but not enough to understand their results. What does it mean that “low AURKB expression” and “high AURKB expression”? How are they classified? What does it mean “according to target gene expression”?

It is still not easy to understand Table 2.

Thanks to the reviewer for pointing this out to us. The expression group of AURKB in Table2 is based on the median expression level of AURKB. We divided all patients into a low expression group (below the median level) and a high expression group (above the median level). Meanwhile, in order to make Table 2 easier to understand, we have made corresponding revisions according to the suggestions of the reviewer, and we would like to thank the reviewer again for your valuable suggestions.

  1. It doesn’t make sense to compare as above. All are just same as comparing between normal and cancer. When there are any differences between gender, age, tumor status, and stages, you can say that. Among pathological stages, AFP levels, and Races, we can see differences of AURKB expression.

Thanks for the suggestions of the reviewers, which is of great significance to the improvement of our article. We have made corresponding modifications in the manuscript according to your suggestions, and the modified part has been marked in red for the convenience of viewing, see for details P7, line 157-165.

  1. It is still tough to understand above. Does “including normal liver tissue from 3 patients, liver tissue from 6 patients with HCC, and liver tissue from 6 patients with Cholangiocarinoma (CHOL).”mean that “including 3 normal liver tissues from 3 HCC patients, 6 HCC tumor tissues from 6 HCC patients, and 6 CHOL tissues from 6 patients with Cholangiocarcinoma.”? Does it mean you have done IHC analysis of 15 tissue samples? I this case, why don’t you show the summary of IHC analysis in table instead of those of just 4 samples?

In general, we show positivity of IHC analysis instead of expression levels.

Immunohistochemical results were directly referenced from THE HUMAN PROTEIN ATLAS (HPA). In this database, protein expression in cancerous tissues is displayed in the Pathology section, so immunohistochemical analysis of these tissue samples was not conducted. The details are included in supplemental table S1.

Thanks to the reviewer who pointed out our shortcomings seriously and responsibly, we have revised the description of IHC analysis results here in response to your suggestion, and the modified part is still marked in red,see for details P3, line 108-111.

  1. I still find wrong data in supplemented original images. How can we trust you? 
  2. Western blotting of knock-down experiment seems to be little bit improved. How many was it repeated? It seems not to be changed growth curve, although western blotting was changed.

After being reminded by the reviewer, we have strictly checked the original data again and found no error. Considering that the uploaded word version may have caused the format problem after the file was uploaded. To avoid this problem we have converted the Original Images for Blots file type to the PDF version. For Western blotting of knock-down experiment, we have carried out three repeats. The WB result of the previous gene knockout test was not so obvious, which may be because the protein AURKB has a relatively long half-life, so we optimized the experimental conditions and performed a new test. The protein collected 72h after transfection (previously collected 48h after infection) was used for WB assay, so there was some improvement in immunoblotting inhibition assay. Does growth curve refer to the result of CCK8? In our study, we examined cell proliferation at 24h, 48h, and 72h, and the results showed significant differences between the knockdown group and the control group (Figure 3C).
